# Perceptions of Stakeholders Regarding China’s Special Education and Inclusive Education Legislation, Law, and Policy: Implications for Student Wellbeing and Mental Health

**DOI:** 10.3390/bs13060515

**Published:** 2023-06-19

**Authors:** Ahmed Alduais, Meng Deng, Hind Alfadda

**Affiliations:** 1Department of Human Sciences, University of Verona, 37129 Verona, Italy; 2Department of English and Communication, The Hong Kong Polytechnic University, Hong Kong, China; 3Department of Special Education, East China Normal University, Shanghai 200062, China; mdeng@ed.ecnu.edu.cn; 4Department of Curriculum and Instruction, King Saud University, Riyadh 11362, Saudi Arabia

**Keywords:** special education, inclusive education, policy, law, legislation, China

## Abstract

Laws and policies, no matter how well designed, can fail if they are not implemented correctly. This can occur when there is no interaction between policymakers and those who are working on the ground. The purpose of this study was to determine the understanding of Chinese stakeholders regarding legislation, policy, and law associated with the provision of special education and explore its implications for student wellbeing and mental health. Two questions were posed: (1) Does a stakeholder’s attitude towards legislation, policy, and law regarding special education impact their role or responsibilities? (2) In what ways do stakeholders interact with legislation, laws, and policies regarding special education and their work experience in the field? Using in-depth interviews as the basis for the study, researchers gained valuable insight into how administrators, practitioners, and academics perceive laws and policies. Participants exhibited exaggerated attitudes and over-interpretations of some of these items, which we attribute to partly real factors, as well as nationalistic or patriotic feelings. The evidence included calls for specific laws and policies as well as a switch from a top-down to a bottom-up approach to reform to bridge the disparity between different regions in the country. As the participants agreed, there have also been remarkable achievements in building a more comprehensive and inclusive system over the last decade. However, the gaps between rural and urban areas, primary and middle schools, high schools and vocational schools need to be addressed urgently in specific laws and policies. Addressing these disparities will not only improve the overall quality of special education but also have significant implications for student wellbeing and mental health. By ensuring that all students have access to tailored support and resources, policymakers can foster a more inclusive and supportive environment that promotes positive mental health outcomes for all learners.

## 1. Introduction

### 1.1. Law, Legislation and Policy

For any policy to be successfully implemented, the interaction between policies, policy-makers, decision-makers, and stakeholders is essential. In the context of special education and inclusive education in China, it is worthwhile to explore the perspectives and experiences of administrators, practitioners, and academics in the field in relation to legislation, law, and policy. 

To gain a better understanding of the status of special and inclusive education policies, we present the following synthesis. Fu et al. mentioned that “in several key government documents on national educational development, the Chinese government persists in developing more special schools” [1] (p. 581) but also raised their concern on “the contradictions in the new regulations reflecting long-rooted disagreements about inclusive education in reality” [1] (p. 581). Further, barriers hindering the development of inclusive education include a competitive exam-based school system, large class sizes (≥45), economic development and more investment, three modes of education (i.e., learning in regular classrooms (LRC), mainstreaming, and home learning), and a paucity of inclusive education in senior middle school (i.e., high school). In order to develop inclusive education, the authors proposed: improving legislation for inclusive education in general, providing detailed laws and policies, identifying the responsible agencies and roles of each in relation to special education and inclusive education, involving the community in legislation processes, repealing laws that are not consistent with the Convention on the Rights of Persons with Disabilities (CRPD), for instance, the law which requires opening special education schools in counties with populations of more than 300,000, as this encouraged segregation and decreased the expansion of inclusion [1].

Moreover, Huang and Zhang conducted a study on the allocation of special education resources in 31 provinces of China (excluding Macau, Hong Kong, and Taiwan) with reference to the National medium and long-term educational reform and development program (2010–2020) and the thirteenth 5 Year Plan for National Education Development [2]. The authors found “Shanghai, Liaoning, Shandong, Jilin, Tianjin, Hebei, Zhejiang, Heilongjiang, Shanxi, and Beijing” are the top provinces in resources allocation of special education as compared to the lowest ones “Sichuan, Chongqing, Fujian, Jiangxi, Guizhou, Guangxi, Qinghai, Yunnan, Hainan, and Anhui”, with the rest 11 provinces in the middle [2] (p. 19). The authors also mentioned that while both Beijing and Shanghai have the advantage of being the most developed provinces, they have the disadvantage of having the largest number of special education students [2]. The authors recommended the issuance of province-based policies and the allocation of resources to bridge the gap in the imbalance among the provinces and provide equal education to people with disabilities, increasing the number of special education schools, the recruitment and preparation of teachers that are more professional, communication among provinces, research on special education and increasing numbers of LRC students [2]. 

Based on the description by authors in reference [2], the first indication of special education appeared in the Book of Rites (i.e., Chinese classics), influenced by Confucian thoughts. Among these is the idea that there must be social harmony, albeit it looked at people emotionally with the view that their disability is due to evil power and ranking them at the lowest social level. The second indicator is the Western missionaries in the mid-19th century, followed by the open reform in 1986. The open reform resulted in major reforms in special education, including the Compulsory Education Law in 1986, moving to inclusive education (i.e., LRC) between 1980 and 1994, the National Plan 2010–2020, and the thirteenth 5 year education development plan [2]. 

To conclude, the levels of legislation in China in relation to inclusive education are hierarchical [1]. The constitutional law did not mention inclusion directly, but its arrangement is discussed within Article 45. The National People’s Congress and its Standing Committee issued several laws that are general and specific. General laws include civil law, criminal law, and administrative law. Specific laws include the Law of Compulsory Education in 1986 on the hosting of special classes and schools and the Law of the People’s Republic of China on the Protection of Persons with Disabilities in 1990. This latter one was amended in 2008 before the ratification of the CRPD. The next level of laws included regulations from education authorities, including Chinese law on education in 1996, mainly Article 10, supporting special education. Regulations made by the State Council have less legal power, and this includes the Regulations Education for Persons with Disabilities in 1994. Further, local provinces’ people’s representatives’ congresses issue ministry regulation and local governmental regulations (e.g., The Shanghai Implementation of the Chinese Law on Protection of Persons with Disabilities. Finally, the ministries and local governments’ regulations like the National Plan 2014 to enhance special education [1].

### 1.2. Rights of Learners with Special Educational Needs

The rights of learners with Special Educational Needs (SENs) are emphasized in Chinese law, legislation, and policies. They strictly inform all education sectors and relevant sectors to ensure *equality*, *learning,* and a *sense of belonging* to all learners with SENs. For instance, the Compulsory Education Law of the People’s Republic of China, Article 19, stated two forms of providing education through special education classes (segregation) or through regular education classes (i.e., LRC) [3]. Another example is the Regulation on the Education of the Disabled details four forms of education for learners with SENs: (a) full inclusion for learners who have mild disabilities, (b) partial inclusion (mainstreaming) for learners who need minor preparation to fully join the regular education system, (c) segregation on special education schools for those who cannot match any of the previous two methods requirements, and (d) exclusion (i.e., home education) for those whose disabilities and conditions do not allow them to join schools on campus [4]. 

Historically, the government has always been highly concerned with ensuring the rights of learners with SENs. The Chinese constitution starting from 1982—revised frequently in 1988, 1993, 1999, and 2004—ensures the protection of people with disabilities. Other important laws include (a) The Law on the Protection of Disabled Persons, 1991, (b) The Rules on the Employment of Disabled Persons, 2007, (c) The Employment Promotion Law, 2007, (d) The Twelfth 5 Year National Programme on Disability (2011–2015), (e) Regulations on Construction of Accessible Environment (2012), and (f) Mental Health Law (2012) (International Labour Organization [ILO], as cited in [5].

Of particular significance, the China Disabled Persons’ Federation (CDPF) was established in 1988 to advocate for the rights and well-being of over 85 million individuals with special educational needs (SENs). As a result, numerous specialized and general associations and organizations have been established to represent this community of learners. These include the State Council Working Committee for Persons with Disabilities, China Association of the Deaf, China Association of the Blind, China Association of the Physically Handicapped, China Association of Persons with Intellectual Disabilities and their Families, China Association of Persons with Mental Disabilities and their Families (ILO) as cited in [5,6,7].

Further, major national policies emphasized ensuring equality, learning, and a sense of belonging for learners with SENs. Among these policies is the National Plan 2010–2020, which was part of the used policy documents in this study—addressing inclusion and segregation [8]. Another yet recent national policy is China Education Modernization 2035 which again emphasized the development of special and inclusive education with more focus on quality [9]. 

The results of the study might be of help to decision-makers when deciding on funding special and inclusive education especially considering bridging the gaps between policymakers and implementers. Policy-makers might also find it useful to use the output of this study, revealing the challenges faced by stakeholders with respect to inclusion and segregation. Practitioners and teachers of special and inclusive education may also find it worth increasing their awareness about the current practices, methods, and theories of special and inclusive education. To this end, the study addressed two questions:Do stakeholders’ attitudes towards legislation, policy, and law regarding special education influence their roles and responsibilities?In what ways do stakeholders interact with the legislation, laws, and policies of special education, and what is their work experience in the field?

## 2. Method

This qualitative, constructivist-based perspective study contributes to understanding special and inclusive education in China through the voices of stakeholders [10,11]. A major benefit of the qualitative approach relevant to this paper is that it studies the context, that is, special and inclusive education in China [12]. The remained of this section discusses the design of the study, data collection, methods, trustworthiness, ethical concerns, and data analysis. 

### 2.1. Study Participants

The study follows the bioecological systems theory in which the examined topic ‘special education development’ is formed, experienced, and understood through the interactions and interrelationships among several different factors within the micro, meso, exo, macro, and chronosystems [13,14,15]. In addition to special education policymakers and practitioners, other stakeholders include special education students, parents, society, the country’s economy, and more. We do not intend to include all these factors in this study. Because of this, the researcher used non-probability sampling, which included both heterogeneity sampling for school principals and practitioners, as well as expert sampling for the 3 academic individuals included in the study. Figure 1 and Table 1 below show the sampling methodology and sociodemographic characteristics of the participants. 

### 2.2. Design

Nine interviews were analyzed using a thematic analysis conducted in 5 phases. It is assumed that school principals, practitioners, and academics have contributed to gaining a deeper understanding by focusing on laws and policies related to the provision of special and inclusive education services in China. Figure 2 illustrates the research design. 

### 2.3. Data Collection

Data was collected between 1 May 2019 and 30 June 2019 in Beijing, China. A total of nine interviews were conducted for this study. Within Beijing, 6 interviews took place, with 3 conducted at a special and inclusive education school and 3 at an intellectual disability school. Furthermore, 2 interviews were conducted at a university, while 1 interview was conducted online. These interviews were between 50 and 75 min long. For all interviews except the online one conducted through WeChat, in which the consent document was verbally signed, mobile phones were used to record the interviews after the consent forms were signed. While 4 academics were scheduled to participate, 1 of them subsequently refused to conduct the interview, stating that he would rather not disclose his views and experiences on the topic. 

Researchers assisting the interviewers were guided by some themes and guiding questions. The form was developed by the researcher, and some of the items and proposed questions were edited by an expert in special education. Using his comments, the prepared list was shortened, and a demographic section was added. During the interviews with nine interviewees, this expert stated only one of them knew English well enough to communicate adequately. Three researchers volunteered to conduct Chinese interviews alongside the researcher. By using WeChat, the researcher arranged and discussed the details of the interview. Some suggestions and questions were raised. A meeting was scheduled to go over their questions 1 by 1 after clarifications were provided to them and their questions answered. We also checked the procedures with 3 assisting researchers who were pursuing their master’s degrees in special education. They were advised, for example, not to discuss policy, laws, and administrative matters of special education in depth with the special education practitioners and to discuss them instead with the director, manager, and principal. In addition, they were instructed not to discuss in detail teaching methods with administrators or too deeply with special education practitioners. Nevertheless, they were told to leave this open so that all interviewees could share their experiences and views regardless of their positions. 

The supervisor approved the interviewees’ list, and the assistants helped to arrange the interview dates and locations. Each interviewee signed a consent form before beginning the interview. Researchers assisting with the interview provided them with Chinese interview questions as well. 

Following the completion of the interviews, 1 interview was first transcribed into Chinese and then translated into English. This transcript was initially deemed too long and stuffed with unnecessary details. Thus, we decided to abstract data according to the questions and themes. Research assistants had the option of transcribing directly into English or starting with Chinese and then translating into English. Nonetheless, the researcher emphasized that strict literal translation should be avoided, and their efforts should be directed toward it. The assisting researchers sent the data to the researcher via WeChat with eight Word documents. This ninth interview was conducted completely online by the researcher, recorded using a mobile phone, and directly transcribed into English. A selective transcription was also used to avoid the inclusion of unnecessary data and waste of time. 

### 2.4. Procedures 

In-depth interviews were conducted with nine special education stakeholders to collect data. As described in Yin (2018), in-depth interviews are known as “unstructured interviews” or “intensive interviews” that are “conversational in nature” (p. 351). A qualitative approach was selected to elicit more data from the interviewees and allow them to speak freely about inclusion and segregation in special education in general (Yin, 2018). The consent form and interview are included in Appendix A, Appendix B, Appendix C. According to Yin (2014), interviews can be used as a valuable tool for validating past data. As he noted, “in-depth interviews have the potential to corroborate established findings and not to expand on topics of a more general, open-ended nature” (p. 179), although gathering more data was the objective of using this approach rather than just confirming quantitative data. 

### 2.5. Trustworthiness

In the present study, we aimed to present a new understanding of special education development in China, with an emphasis on presenting a critical analysis of the conflict between policies and stakeholders. In conducting a thematic analysis for the collection and analysis of the interviews, there were five phases. As part of this, we compiled data, disassembled, reassembled, interpreted, and concluded. 

The nine interviews were analyzed primarily using content analysis [16,17]. A thematic analysis is considered trustworthy through the different phases of the process [18,19]. For this study, it was established through various methods. Credibility, transferability, dependability, and confirmability are the four main trustworthiness criteria, each of which was approached differently during each of the 5 phases of thematic analysis, including compiling, dissembling, reassembling, interpreting, and concluding. Each of these phases, the trustworthiness concepts, and the means that were used to establish them are listed in Table 2.

### 2.6. Ethical Concerns 

During the in-depth interviews, agreements were arranged at the supervisor’s end since many interviewees did not possess a good level of English, which would have allowed them to conduct the interviews in English. Before beginning the interview, all interviewees signed a consent form that was located in the Appendix B. In the ninth interview, the consent form was approved verbally, and the interview was conducted online. 

We also ensured the conformation to ethical standards during the interviews, where visited schools requested that photographs not be taken, and the names of schools and public universities were not revealed. In accordance with the interviewees’ requests, their names and affiliations were also coded. 

### 2.7. Data Analysis 

The interviews were analyzed following 5 phases: compilation, disassembly, reassembly, interpretation, and conclusion (see Appendix A, Appendix B, Appendix C) for data coding, consent form and interview protocol. Once the data were compiled, disassembled, and reassembled, it was arranged based on the views and experiences of leaders, practitioners, and academics. Following the themes which were listed on the interview form, the rest of the analysis followed (see Figure 3 and Figure 4). We selected excerpts based on their relevance to the themes as well as summarizing and paraphrasing them. When discussing the results, discussing the discussion, and discussing the rest of the study, the interviewees were referred to by their titles (director, principal, manager, special education practitioner, and special education academic/expert). We replaced the names of the schools and the university affiliations of the interviewees with (special education school, intellectual disability school, university in Beijing, university in Wuhan, and university in southwest China). In instances where data corresponding to the desired theme was absent, the term “not available” (NA) was entered into the data extraction tables. 

## 3. Results 

### 3.1. Stakeholders’ Views on Policy and Law of Special Education 

#### Result Overview

While nine questions guided the interview (see Appendix C), there was also room for open discussion within these topics. Two of the questions pertain to a personal story describing the participants’ attitudes toward their motivation to work in special education. It was stated by two of the administrators that they had moved from teaching into administration versus the Director and Principal, who had 39 years of experience. Three practitioners reported their reason as the major which is similar to those academics who reported choosing special education as it was challenging and they were able to have a better job, except for the assistant professor with two years’ experience who reported being compelled to accomplish this goal.

All of the remaining questions were designed to gain an understanding of special education law and policy in China. The topic is further divided into two parts. This first part looks at awareness of special education in China and attitudes toward it. The second section discusses the National Plan 2010–2020 as it pertains to the four main instructions contained in this policy document: (1) caring for and supporting special education, (2) improving special education, (3) perfecting guarantee access to special education, and (4) developing special education. 

The analysis of the policy and law regarding special education revealed several findings. While it is true that some special education personnel may hold excessively positive attitudes towards special education policies and laws, it is crucial to emphasize the importance of developing more effective and targeted laws and policies that are specific to each province, as well as to special education and inclusive education. It is essential that these laws and policies are accurately implemented, taking into consideration the fact that the awareness of special education personnel regarding special education laws and policies is often average or below average. Additionally, it is worth noting that some special education personnel may exhibit an exaggerated negative attitude towards these policies and laws, while others may tend to over-interpret the existing laws and policies in the field of special and inclusive education.

### 3.2. Attitudes toward Inclusion and Segregation 

**Exaggerated Attitudes**. We use this concept to refer to an overemphasis or overstatement of one’s opinions or beliefs, often leading to an inflated or extreme view of a particular subject or issue. That being said, it is imperative to prioritize the development and implementation of more effective laws and policies that are tailored to the specific needs of each province, as well as to the domains of special education and inclusive education. This is crucial due to the observation that the awareness of special education personnel regarding special education laws and policies tends to be either average or below average. By issuing and accurately implementing province-specific, special education-specific, and inclusive education-specific laws and policies, we can address this gap in knowledge and ensure more effective practices in the field.

Some of the participants mentioned the below laws and policies, while some others stated that they could not recall any: 


*Regulations on the education of the disabled … law on the protection of disabled persons*
*…law on the protection of minors … compulsory education law … amendment of special education policy in 2017 … National Plan 2010–2020 … disability protection…. (SES-1; IDS-1; SES-2; IDS-2; SES-3; IDS-3).*


**Negative Attitudes**. Additionally, special education personnel have exaggerated negative attitudes toward special education policies and laws. 

*Should popularize laws and policies … good policies, but need to be enforced … not specific laws but in progress … need to follow bottom-up approach … to ensure all personnel and disabled rights … have general laws, policies, and legislations … good regulations and laws but implementation is not taking place in full … lack of specific and detailed laws on special education (PU-1; PU-2; PU-3)*.

**Overinterpretations of Laws**. We use this concept to refer to the act of assigning excessive meaning, significance, or implications to legal provisions, often leading to an exaggerated or distorted understanding of their intended scope or application. Based on this, it is essential to issue and accurately implement more effective laws and policies that are tailored to specific provinces, special education, and inclusive education. This is necessary because certain individuals in the field of special and inclusive education tend to over-interpret the existing laws and policies. By developing and implementing province-specific, special education-specific, and inclusive education-specific laws and policies, we can ensure clearer guidelines and avoid misinterpretations that may hinder the effective implementation of these regulations.


*Integration laws at the province, city, and district levels in Beijing [do exist] … all are improving well … improving but still needs further policies … local and national developments are working well … remarkable progress but still needs more efforts … effective development … moving towards zero rejection (SES-1; IDS-1; SES-2; IDS-2; SES-3; IDS-3; PU-1; PU-2; PU-3).*


Hence, although special education personnel have displayed exaggerated positive attitudes towards special education policies and laws, it is crucial to develop and implement more precise laws and policies that are tailored to specific provinces, special education, and inclusive education. In this regard, Figure 5 provides a visual representation of stakeholders’ awareness, reflections, and perspectives on laws and provisions.

### 3.3. Reflections on Inclusion and Segregation in the National Plan 2010–2020 

The analysis of policy and law of special education also put more focus on the National Plan 2010–2020. Although the deadline for achieving the National Plan 2010–2020 was approaching when this study was conducted, some of the acts have not been put into effect at the national level, especially in rural areas, as evidenced by the expressed concerns of some of the participants, the National Plan 2010–2020 has been introduced to ensure both quantitative and qualitative development of special education in China because it aims at caring for and supporting special education, it considers improving the special education system, it ensures perfecting guarantees for special education, and it emphasizes developing special education. 

**Caring for and Supporting Special Education**. First, the National Plan 2010–2020 has been introduced to ensure both quantitative and qualitative development of special education in China because it aims at caring for and supporting special education where government at all levels should speed up special education development, intensive training for people with disabilities, and full integration into the society. The reflection of the participants on this particular article for reforming special education included: 


*Gradually implemented … providing personalized services representing the country’s development of economy, culture and civilization … a lot of special education policy support and some welfare support, not only for the children’s education but also for the children’s life … supervisors and managers need to learn more but not practitioners … increase in the investment including financial and policy support including teacher development and extending the 9 year compulsory education to high school and vocational education … the country’s attitude towards special education has undergone great changes, from promotion to support to success … has limited impact but a good start. (SES-1; IDS-1; SES-2; IDS-2; SES-3; IDS-3; PU-1; PU-2; PU-3).*


**Improving Special Education System**. Additionally, it considers improving the special education system proposing that a county with over 300,000 with disabilities should have a special education school; schools create opportunities to accept them in regular education and preschool education, senior middle school, vocational education, and higher education are accelerated. Some of the interviewees showed total agreement that these are clearly observed, and they declared: 


*There were no special schools in many places, but now there are 2205 special schools … in the past 5 or 6 years, the Haidian district has paid more and more attention to integrated education … more and more special education schools are opening; before 2010, very few schools were providing special education for vocational education (SES-1; IDS-1; SES-2; IDS-2; SES-3; IDS-3; PU-1; PU-2; PU-3).*


Nevertheless, some others think that the realization of this Article is still partial and claim: 


*Although there are regulations in the policies, it is difficult for kindergartens to accept special education children … the policy of setting up schools based on population is not fair … high school, vocational education and higher education still need too much to be done. (SES-1; IDS-1; SES-2; IDS-2; SES-3; IDS-3; PU-1; PU-2; PU-3).*


**Perfecting Guarantees for Special Education**. Furthermore, it ensures perfecting guarantees for special education by making basic national standards for the operation of special education set by the local government, more investment, ensuring that regular schools provide a suitable environment, increasing the salaries of special education teachers, funding special education in the community (hard situation families), and providing free senior middle school gradually. The views on this Article are mixed between crediting its full realization and the need to issue further detailed policies and laws to ensure the best special education services. 


*In the implementation of Beijing students’ food costs, clothing costs are gradually met … this is being noticed but still needs to be more effective … preschool education and vocational education related to special children, especially those with mental retardation, are very weak … in some areas, they do not even have a high school within their system … the basic salary for them should be 30% more than regular teachers … more policy documents need to be issued in the next 10 years … the gap between regions, unbalanced investment and spending, and the economic status of families need urgent and further reform. (SES-1; IDS-1; SES-2; IDS-2; SES-3; IDS-3; PU-1; PU-2; PU-3).*


**Comprehensive Development**. Finally, the National Plan 2010–2020 has been introduced to ensure both quantitative and qualitative development of special education in China because it emphasizes developing special education where schools are refurbished, expanded, or new ones built, education conditions should have necessary teaching, living, rehabilitation training facilities, and teachers have professional training. Positive views included: 


*The standards for schools’ construction were set in 2011, and in Beijing in 2012 … with more development and more foreign exchanges, we can do better … hoping to see further development in teacher training … inequalities among different areas … … too many professional training programs for special education teachers, including those in schools, districts, and provinces (SES-1; IDS-1; SES-2; IDS-2; SES-3; IDS-3; PU-1; PU-2; PU-3).*


Dissimilar to such positive reflections, some others think: 


*Lots of schools in poor areas are rebuilt, and some are refurbished so that teachers can have more access … within regions such as Beijing, there are a lot of schools that have different levels of development, education is far worse, and nationally, the imbalance is bigger (SES-1; IDS-1; SES-2; IDS-2; SES-3; IDS-3; PU-1; PU-2; PU-3).*


Hence, while the deadline for achieving the National Plan 2010–2020 was approaching when this study was conducted, some of the acts have not been put into effect at the national level, especially in rural areas, as evidenced by the expressed concerns of some of the participants, the National Plan 2010–2020 has been introduced to ensure both quantitative and qualitative development of special education in China. Figure 6 below is a visualization of the stakeholders’ understanding and views on the National Plan 2010–2020 Articles related to special and inclusive education. 

## 4. Discussion

The purpose of this study was to uncover how Chinese stakeholders perceive laws and policies regarding special education and inclusive education throughout the country. These findings reveal the possible gaps between policy-makers and implementers (i.e., stakeholders) and how this gap may affect the system’s outcome. The findings may be broken into two parts. In the first part, the following findings were found: (1) academics and administrators have a deeper understanding of and familiarity with laws and policies than practitioners; (2) the included stakeholders have exaggerated views and interpretations of the current laws and policies; and (3) while some of the included stakeholders think the provision is moving forward despite a few shortcomings, others believe that development should move to a bottom-up approach and more specific policies and laws should be issued to developing and underdeveloped areas.

The second section of the findings includes the following: (4) in regard to promoting special education, it appears that the government has been shifting from promotion to real support and achievement, although this remains limited in rural areas, (5) when it comes to improving special education, there are also some signs of progress between 2010 and 2020. However, special education services at the high school, vocational school, and preschool levels remain insufficient, (6) For the development of special education standards, it is believed that these standards move from the top to the bottom, which takes some time to reach the remote areas in a large country such as China, and (7) for perfecting the special education system, they believe this is focused on major provinces and cities such as Beijing and Shanghai, and there is no investment in other regions across the country.

It is clear from these findings that China has made significant progress in its special education system, but there are still several issues and issues that need to be addressed to provide better services for the Special Education community [20,21,22]. There have also been reports of the narrow definition of special education in China [23], the gap between rural and urban areas [24], social justice and special education services [25], social support for teachers [26], and curriculum relevant to specific categories of this community [27].

Two interpretations are possible based on these findings. It is apparent from the understanding and reflections of the included stakeholders that there are exaggerated attitudes toward the development of special and inclusive education in China. Compared to the rest of the world, these views included either negative attitudes asserting development is slow and restricted to large cities and provinces or positive attitudes asserting that the current situation represents the best of the available options. There is a possibility that these views can be attributed to an oversimplification of laws and policies as well as nationalist views. But what is actually evident is that there is significant progress and a successful special education system across the country regardless of the gaps in the different regions.

The National Plan for the Development of Education, which includes special education, has had a profound effect on the quality and quantity of services provided across the country. Evidence included opposing views that this development occurred only in basic and middle schools but not in preschool, high school, vocational schools, or universities. 

Several implications can be drawn from the above interpretation. First, the understanding of laws and policies by stakeholders influence their interaction with their working environment (e.g., schools, learners, society) and, as a result, either facilitates or hinders the performance of their work. The stakeholders within our study demonstrated high levels of enthusiasm and positive attitudes toward the practicality of the policies and laws being issued and implemented in China. There is no doubt that they express some negative views regarding the discrepancy between regions, and they place greater emphasis on certain levels (basic schools and middle schools) over others (i.e., preschools, high schools, vocational schools, and higher education).

As a result, it has been confirmed that the National Plan 2010–2020 has been successful, indicating that all stakeholders and, hopefully, the special education community, parents, and society as a whole, are aware of the enhancement of services provided in the fields of special education and inclusive education. While this affirmation includes several limitations on such policies, these limitations should serve to show the government why it should switch to a bottom-up approach (i.e., more budget and spending for underdeveloped or developing regions) rather than a top-down approach.

The findings and interpretations of this study have significant implications for student wellbeing and mental health. Ensuring that special education services are comprehensive, inclusive, and accessible to all students, regardless of their geographical location or level of education, is crucial for fostering a supportive environment that promotes positive mental health outcomes for students with diverse learning needs.

Addressing the gaps in understanding and implementation of laws and policies among stakeholders can lead to a more equitable and effective special education system, ultimately benefiting students’ wellbeing. By enhancing the quality and accessibility of special education services across various regions and educational levels, students with special needs will receive the necessary support for their mental and emotional well-being. Furthermore, a bottom-up approach that emphasizes investment in underdeveloped or developing regions can lead to more equitable access to services, reducing the disparities in mental health support for students with special needs. This approach acknowledges the importance of a comprehensive and cohesive special education system in promoting social justice and fostering a sense of belonging and wellbeing for all students, regardless of their unique learning needs.

Lastly, fostering open communication and collaboration among all stakeholders involved in special education and inclusive education can have a positive impact on student wellbeing and mental health. When administrators, educators, parents, and policymakers work together to better understand and implement laws and policies, they create an environment that is more supportive and responsive to the unique needs of students with disabilities. This collaborative approach helps ensure that special education services are tailored to students’ individual needs, promoting their overall wellbeing and mental health. Furthermore, increased awareness and understanding of the challenges faced by students with special needs can contribute to a more empathetic and inclusive educational environment, further enhancing their mental health and sense of belonging within their school communities.

## 5. Limitations

The present study aimed to reveal a more in-depth understanding of Chinese stakeholders regarding the laws and policies regarding special and inclusive education. There are, however, two limitations in the study. Firstly, the collected data was limited to three categories of stakeholders (administrators, academics, and practitioners). Due to linguistic and administrative barriers, it was not possible to gain access to other stakeholders. Secondly, the stakeholders involved are mainly from developed areas in China, and their views may fall in line with the developed environment in which they work. 

Another potential limitation of the study is the relatively small sample size, consisting of only 9 interviews. However, it is important to note that these interviews were conducted in-depth, allowing for rich and detailed insights from participants. Despite the limited number, the depth of the interviews can provide valuable information for understanding stakeholders’ perspectives.

Furthermore, it should be acknowledged that the majority of the interviews were conducted in Beijing, which is the location of the central government and known for its advanced development. This geographic concentration may introduce some bias in the data, as it may not fully capture the experiences and perspectives of stakeholders in underdeveloped or developing regions across the vast country of China. It is important to consider the potential limitations of generalizing the findings to the entire country.

We acknowledge these limitations and their potential impact on the generalizability of our findings. Future studies could aim to include a larger and more diverse sample, incorporating perspectives from stakeholders across various regions in China to provide a more comprehensive understanding of the issues related to special education and inclusive education legislation, law, and policy.

## 6. Conclusions 

In this study, we sought to explore Chinese stakeholders’ understandings and experiences of special education policies and laws. In our findings, the attitudes of stakeholders and their interaction with policy play a significant role in the effectiveness of their work and in the implementation of policy. Participants showed both positive and negative attitudes toward certain laws and policies. The major concerns are the gaps in development, budget allocations, and reforms across the country. Although national policies are issued, their implementation is left up to each school, city, and province, and this is where gaps emerge, whether they arise with regard to policies and their implementation or service provision across the country or even between one school and another within a city.

## Figures and Tables

**Figure 1 behavsci-13-00515-f001:**
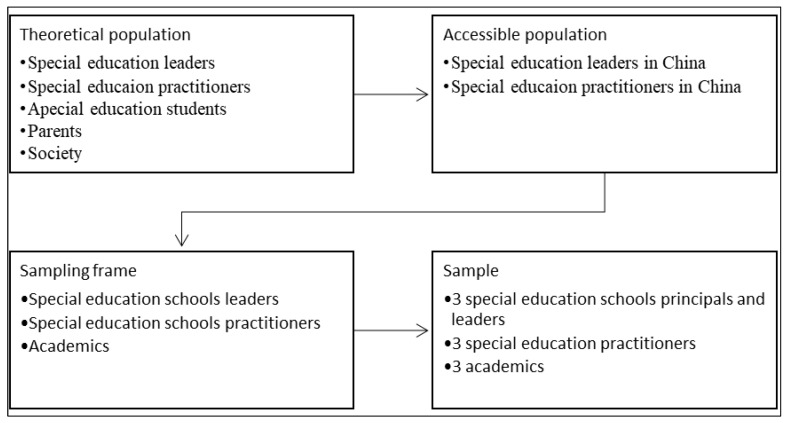
Sampling Framework.

**Figure 2 behavsci-13-00515-f002:**
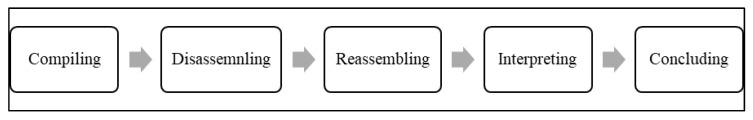
Thematic Analysis Study Design.

**Figure 3 behavsci-13-00515-f003:**
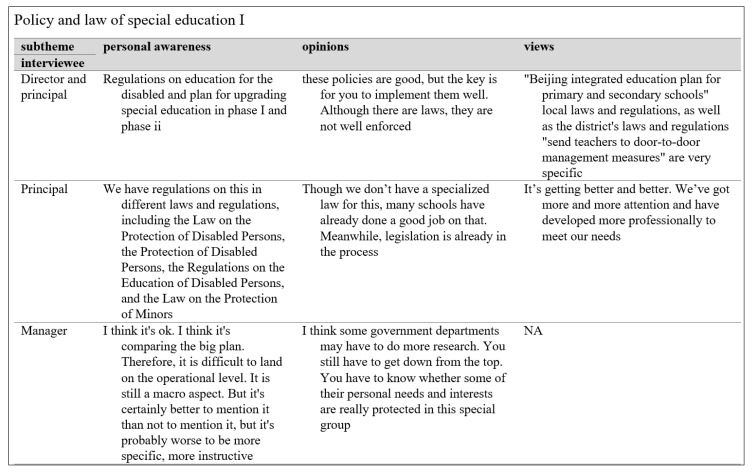
Sample Data Extraction Screenshot for Stakeholders’ Understanding and Views on Laws and Policies.

**Figure 4 behavsci-13-00515-f004:**
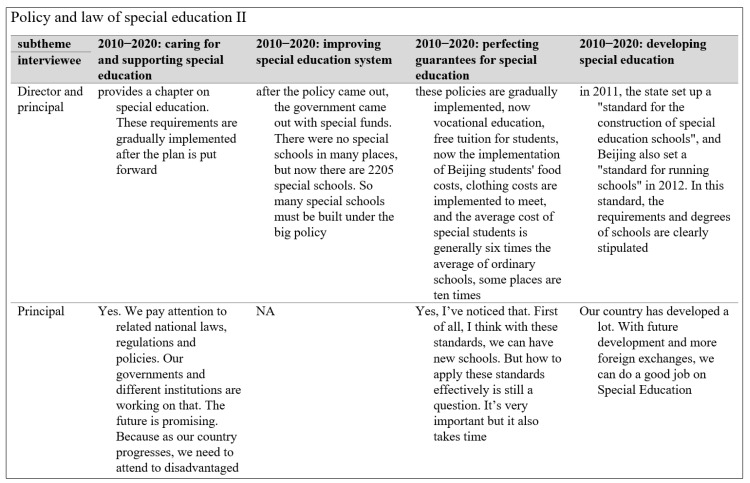
Sample Data Extraction Screenshot for Stakeholders’ Views on the National Plan 2010–2020.

**Figure 5 behavsci-13-00515-f005:**
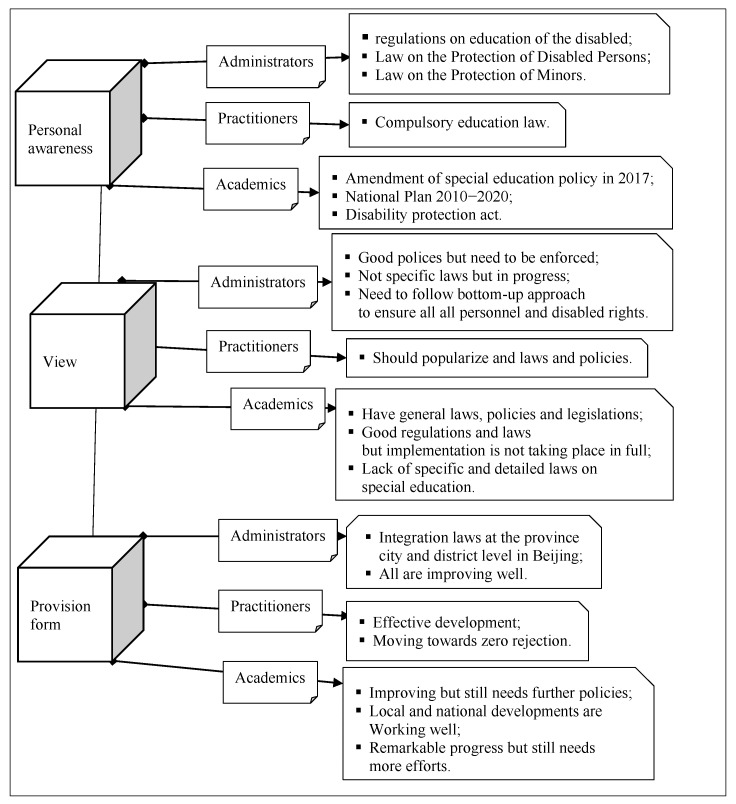
Policy and Law of Special Education I.

**Figure 6 behavsci-13-00515-f006:**
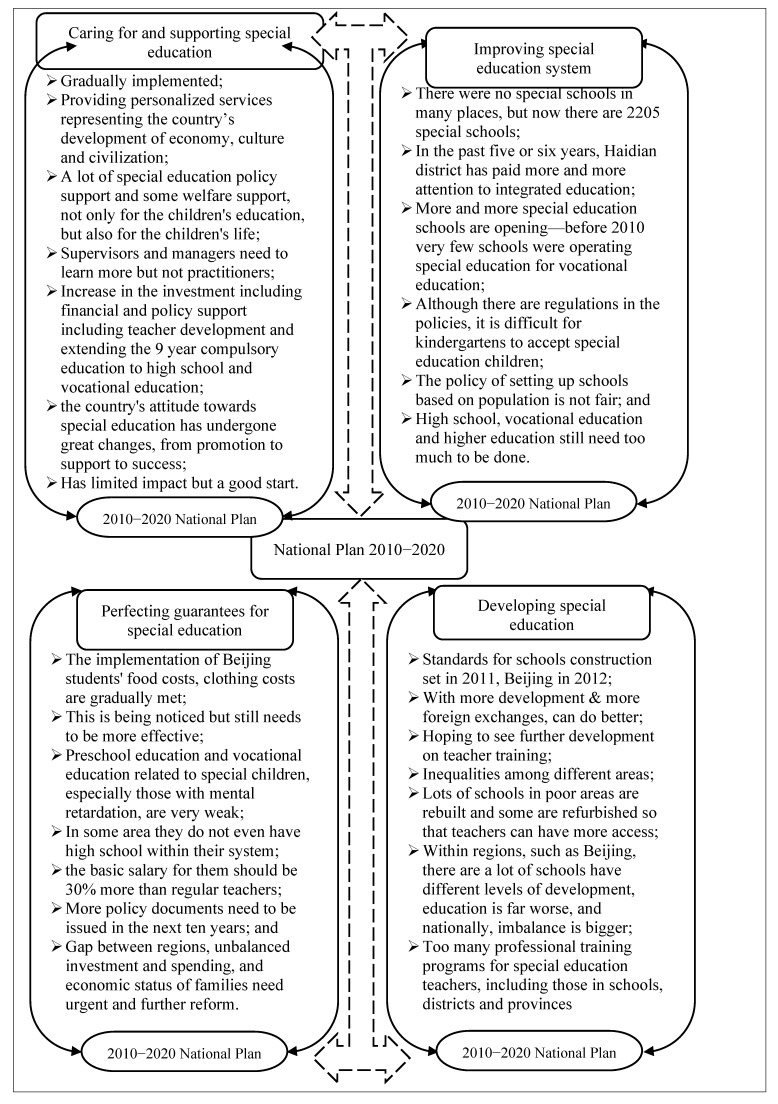
Policy and Law of Special Education II.

**Table 1 behavsci-13-00515-t001:** Sociodemographic Characteristics of the Participants.

Position	Age	Gender	Experience	Major	Location	Institution	Pseudonym	Code
Director and principal	60	Male	39	Teacher Education	Beijing	Special Education School	A special education school director	SES-1
Principal	52	Female	31	Special Education	Beijing	Intellectual Disability School	A special education school principal	IDS-1
Manager	35	Female	14	Special Education	Beijing	Special Education School	A special education school manager	SES-2
Practitioner	38	Male	16	Special Education	Beijing	Intellectual Disability School	A special education school practitioner	IDS-2
Practitioner	33	Female	7	Special Education	Beijing	Special Education School	A special education school practitioner	SES-3
Practitioner	30	Female	3	Primary Education	Beijing	Intellectual Disability School	A special education school practitioner	IDS-3
Assistant Professor	33	Male	11	Special Education	Wuhan	Public university	An academic	PU-1
Postdoctoral fellow	29	Male	7	Special Education	Beijing	Public university	An academic	PU-2
Assistant Professor	30	Male	2	Special Education	Chongqing	Public university	An academic	PU-3

**Table 2 behavsci-13-00515-t002:** Trustworthiness Establishment at the Five Phases of Thematic Analysis.

Phase	Concepts	Means	Explanation
Compiling	Credibility	Triangulation	The study used in-depth interviews; however, these interviews were conducted under the guidance of some guiding themes and questions that enabled the verification of the collected data from nine interviews.
	Transferability	Thick description	The study presents a detailed description of data compilation in its methods section.
	Dependability	Detailed documentation	This study provides detailed instructions on how to compile data from the procedure section.
	Confirmability	Peer checking	Researchers in special education verified the data and, further, the researcher himself verified the data.
Disassembling	Credibility	Triangulation	An in-depth interview generated a large amount of data. It took a long time to transcript the first interview, which led to thick data. Consequently, the data was selectively transcribed. The process of transcribing everything and then translating everything into Chinese was incredibly time-consuming. To perform the first step of verification for the abstracted data, we selected one of the three researchers who were best at English.
	Transferability	Thick description	An in-depth description of coding and analysis can be found in the procedure and data analysis section.
	Dependability	Detailed documentation	A complete description of the disassembling process was provided.
	Confirmability	Peer checking	As part of the three researchers helping with the interview, transcription, and translation into English, one of them also checked the coding of themes and analysis steps.
Reassembling	Credibility	The emic or folk perspectives of the participants	Researchers who contributed to data collection were instructed to avoid any changes to the questions and to avoid leading the interviews in any direction. In addition, they were informed to give the interviewees full freedom to share their opinions and experiences without pressuring them toward the interviewers’ or researchers’ views. Moreover, the researcher requested that researchers transfer the data honestly and literally and avoid creative translation by all means.
	Transferability	Thick description	A detailed description of the final themes is provided on the data analysis page.
	Dependability	Detailed documentation	A detailed description of all steps, processes, and procedures is provided.
	Confirmability	Peer checking	For the purposes of ensuring that the final data is not beyond the scope of the study, both the generated and emerging themes were compared to the study’s objectives. The comparison was verified by a peer in the same field.
Interpreting	Credibility	Progressive subjectivity checks	Occasionally, the researcher checked if the data analyzed supported some proposed theses or contradicted others. This has been left and further discussed in the study discussion, limitations, and future research.
	Transferability	Thick description	A detailed explanation of the interpretation is provided in the procedure section and data analysis section.
	Dependability	Detailed documentation	The steps, processes, and procedures of interpretation are documented in detail.
	Confirmability	Repeated checking	The interpretation is then verified and confirmed through repeated checks.
Concluding	Credibility	peer debriefing	Peers suggested going straight to the points with the conclusions and their relevance and value to the field, society, and country.
	Transferability	Thick description	For results, analysis and interpretation were used. For conclusions, discussion and inference were used.
	Dependability	Detailed documentation	Detailed documentation is available on all steps, processes, and procedures involved in concluding data.
	Confirmability	Peer checking	The conclusions were peer-reviewed and are based on the findings and yet still answer the research questions.

## Data Availability

The data presented in this study are available on request from the first author.

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
