# Peer review of "Perceptions of Stakeholders Regarding China’s Special Education and Inclusive Education Legislation, Law, and Policy: Implications for Student Wellbeing and Mental Health"

_behavsci, 2023, doi:10.3390/bs13060515_

Round 1
Reviewer 1 Report
The study is of scientific interest and has rigor in its presentation. Some aspects should be improved.
1. Introduction
Line 58: first time the acronym LRC appears. It is a central concept in the text, so it is convenient to put its meaning and not just the acronym.
2. Data Collection
Line 11: “total of six interviews were conducted in Beijing”. It’s correct? You have a total of nine interviewees.
3. Procedures
Lines 62-64: “(see 3.6 for more details on the procedure, duration, themes, etc.) (...). The consent form and interview are included in Appendix B.” The article doesn’t have 3.6., nor Appendix B.
4. Discussion
Suggestion: cross-check with the predicted in China Education Modernization 2035 (2019)
4. Tables and Figures
Review the framing in the text:
(a) avoid overlapping information;
b) avoid overly long tables;
c) improve formatting/framework.
2. References
a) The following reference is missing in the final: Fu et al. (2019)
b) The author uses, in the text, the system (author, date) and not [number]
Reviewer 2 Report
The purpose of the study was to determine Chinese stakeholders' understanding of legislation, policy, and law related to the provision of special education. While the topic is important and the manuscript is overall well written, several major concerns need to be addressed before further evaluation is considered.
Major Concerns:
- The authors emphasize "student well-being" and "mental health" in the title, but neither is mentioned much in the main text.
- I don't think the two questions raised at the end of the Introduction are well addressed in the rest of the paper. Specifically, the manuscript does not answer how stakeholders' attitudes "influence their roles and responsibilities" (question 1), or how stakeholders "interact" with legislation, laws, and policies (question 2).
- The sample size is very small and highly biased (7/9 participants are from Beijing). This limitation should be fully discussed.
- It is unclear which nine questions guided the interview. (Line 123)
- There are multiple problems in the results section.
(1) The same sentense “more effective laws and policies that are province specific, special education specific and inclusive education specific should be issued and implemented accurately because the awareness of special education personnel of special education laws and polices is either average or below average” is repeated four times around line 141-180, and it is not fully supported by the quoted responses.
(2) What do "exaggerated attitudes" (line 152) and "overinterpretation of laws" (line 169) mean? What attitude is exaggerated and what aspect of the laws is overinterpreted?
(3) Are there obvious inconsistencies between the number of quotations and the number of participants (line 227, line 233, line 262, line 268, etc.)? - Line 349-350: it is unclear why bottom-up approach is interpreted as spending budget for underdeveloped or developing regions.
Minor points:
- Line 58: add the full name of LRC
- Line 115: add the full name of SENs
- Line 141-150: this paragraph should be rewritten.
English language is fine.
Round 2
Reviewer 2 Report
The revised manuscript addresses most of my concerns. I have only one additional comment regarding my previous point #1. If the authors insist on including "Student Wellbeing and Mental Health" in the title, they should explicitly introduce and discuss how their findings relate to student wellbeing and mental health.
Author Response
Dear Colleague,
Thank you for your valuable feedback and for helping us to improve the quality of our manuscript. We appreciate your time and effort in reviewing our work.
In response to your comment regarding the inclusion of "Student Wellbeing and Mental Health" in the title, we have added three paragraphs in the discussion section of the revised manuscript. These paragraphs explicitly introduce and discuss the relevance of our findings to students' mental health and wellbeing. We believe that this addition provides a more comprehensive understanding of how the findings and interpretations of our study can have significant implications for student wellbeing and mental health in the context of special education and inclusive education policies.
We hope that the revisions made to the manuscript address your concerns effectively, and we are grateful for your guidance in enhancing the overall quality of our work.
Sincerely,
Authors